# Morphometric Analysis Reveals New Data in the History of *Vitis* Cultivars

**DOI:** 10.3390/plants14162481

**Published:** 2025-08-10

**Authors:** José Javier Martín-Gómez, José Luis Rodríguez-Lorenzo, Francisco Emmanuel Espinosa-Roldán, Félix Cabello Sáenz de Santamaría, Gregorio Muñoz-Organero, Ángel Tocino, Emilio Cervantes

**Affiliations:** 1Instituto de Recursos Naturales y Agrobiología de Salamanca (IRNASA), Consejo Superior de Investigaciones Científicas (CSIC), Cordel de Merinas, 40, 37008 Salamanca, Spain; jjavier.martin@irnasa.csic.es; 2Plant Developmental Genetics, Institute of Biophysics v.v.i, Academy of Sciences of the Czech Republic, Královopolská 135, 612 65 Brno, Czech Republic; rodriguez@ibp.cz; 3Instituto Madrileño de Investigación y Desarrollo Rural Agrario y Alimentario (IMIDRA), Finca El Encín, 28805 Alcalá de Henares, Spain; franciscoemmanuel.espinosa@madrid.org (F.E.E.-R.); felix.cabello@madrid.org (F.C.S.d.S.); gregorio.munoz@madrid.org (G.M.-O.); 4Departamento de Matemáticas, Facultad de Ciencias, Universidad de Salamanca, Plaza de la Merced 1-4, 37008 Salamanca, Spain; bacon@usal.es

**Keywords:** curvature, grape diversity, morphometric geometry, shape analysis, solidity, *Vitis* cultivars

## Abstract

Seeds of different *Vitis* cultivars (*V. vinifera* subsp. *vinifera*) have an interesting diversity of shapes, ranging from the small seeds of high solidity and low aspect ratio in some species of *Vitis* and *V. vinifera* subsp. *Sylvestris* to other morphological types with elongated stalks, characteristic of the more recent cultivars, suggesting a transition with alterations in seed shape associated with groups of cultivars. *J*-index analysis is a morphometrical technique that measures the percentage similarity of seed images with geometric models. Three models based on the outlines of reference cultivars (a model based on the Spanish female cultivar Hebén; and mixed models for French and German Chenin and Gewürtztraminer, both related to Savagnin Blanc; and Regina dei Vigneti and Muscat Hamburg, related with the Muscat group) have been applied to select the average outlines (Aos) resembling these models from a collection of cultivars maintained at IMIDRA. Three groups resulted, called Hebén, Chenin, and Regina, with 15, 25, and 18 cultivars, respectively. Principal component analysis (PCA) with the Fourier coefficients of the Aos for these cultivars and seeds of other species of *Vitis* and *V. vinifera* subsp. *sylvestris* showed differences between groups. Specific Fourier coefficients were related with geometric properties of the seeds, circularity, roundness, aspect ratio, and solidity as well as with diverse measurements of curvature allowing to establish hypothesis about the change in geometric properties along the evolution of cultivars.

## 1. Introduction

Viticulture is associated with some of the earliest known civilizations. Traces of human activity alongside remains of grape seeds have been found dating back to the Neolithic period in the southern Caucasus [1], the Mediterranean region [2,3,4], and other areas of Eurasia and Africa [5,6,7]. Since then, major changes have often resulted from husbandry, particularly the domestication of *Vitis sylvestris* grapes or the crossing of existing varieties [8]. Cultivation of *Vitis* modifies the reproductive type from dioecious in wild populations to hermaphroditic in cultivation, alongside improved yield and organoleptic properties. Throughout history, some female cultivars have been proven highly valuable as parents in crosses for creating new varieties, such as Hebén [9,10]. Today, grapevine cultivars exhibit high diversity, with a variation exceeding that observed in sylvestris populations [11,12,13]. This diversity is also reflected in the wide variety of seed shapes. In addition to their undoubted economic value, grapes are an interesting model for studying biodiversity and have great potential for future development, particularly regarding the application of new technologies to their classification and phylogeny.

Thanks to advances in genomics, the study of *Vitis* has changed significantly over the past two decades. Grapes have a long life cycle and a complex genome comprising 19 chromosomes (2n = 38), which makes the development of classical genetic maps based on crosses with parental plants exhibiting different phenotypic traits difficult, unlike in *Drosophila* or *Arabidopsis* [14,15]. Nevertheless, the development of DNA amplification by polymerase chain reaction (PCR) has enabled the identification of genetic markers [16,17,18,19] and the creation of genetic maps [20,21,22,23,24]. Genome sequencing has provided additional markers, as well as new tools with which to understand the relationships between cultivars.

Today, there are several thousand grape cultivars worldwide, most of which are the result of crosses involving ancient cultivars from diverse regions such as Africa, Asia, the Caucasus, the Iberian Peninsula, and Central Europe [9,10,11,12,13,25]. Whole-genome sequencing (WGS) projects have identified the genotypes corresponding to the genetic pools corresponding to different *Vitis* ancestors of modern cultivars, emphasizing some that may represent ancient haplotypes responsible for the genetic constitution underlying the modern cultivars [25]. However, the significant increase in genomic resources and data has not always been accompanied by progress in understanding variation in simple traits. For example, progress in understanding variations in seed morphology has been slower.

For a long time, seed morphology has formed the basis of taxonomy in many plant families [26]. The Vitaceae exhibit an interesting diversity in seed shape [27], with similarities to geometric forms such as heart, pear, and water drop curves [28]. Differences in seed shape have been found between sylvestris and cultivated varieties, with larger seeds in cultivated species (*V. vinifera* L. subsp. *vinifera*) and more variable in size than those of wild species (*V. amurensis* Rupr., *V. californica* Benth., *V. labrusca* L., *V. aestivalis* Michx.) [29,30,31]. Since the work of Terral et al. [30], studies on seed shape diversity have often been based on analyses of seed outlines using Elliptic Fourier Transform (EFT) equations.

The parametric functions representing a closed-plane curve may be approximated by sums of trigonometric functions as truncated Fourier expansions [32,33]. When the parametric components are piecewise linear functions $x(t)=\sum_{i=1}^{K}\Delta x_{i}$, $y(t)=\sum_{i=1}^{K}\Delta y_{i}$, these expansions are:(1)$$X_{N}\left(t\right)=A_{0}+\sum_{n=1}^{N}a_{n}\cos\left(\frac{2n\pi t}{T}\right)+b_{n}\sin\left(\frac{2n\pi t}{T}\right)$$(2)$$Y_{N}\left(t\right)=C_{0}+\sum_{n=1}^{N}c_{n}\cos\left(\frac{2n\pi t}{T}\right)+d_{n}\sin\left(\frac{2n\pi t}{T}\right)$$
where *i* represents the index of summation for the piecewise linear segments that approximate the parametric components *(x(t))* and *(y(t)), K* represents the index of summation over the *(K)* piecewise linear segments when computing the Fourier coefficients, and *n* represents the index of the Fourier harmonics (or frequency components) in the truncated Fourier series, and $A_{0}$ and $C_{0}$ define the mean value of each contour component, and the coefficients $a_{n},b_{n},c_{n},d_{n}$ are calculated by the equations:(3)$$a_{N}=\frac{T}{2\pi^{2}n^{2}}\sum_{p=1}^{K}\frac{∆x_{p}}{∆t_{p}}\cos\left(\frac{2n\pi t_{p}}{T}\right)-\cos\left(\frac{2n\pi t_{p-1}}{T}\right),$$(4)$$b_{N}=\frac{T}{2\pi^{2}n^{2}}\sum_{p=1}^{K}\frac{∆x_{p}}{∆t_{p}}\sin\left(\frac{2n\pi t_{p}}{T}\right)-\sin\left(\frac{2n\pi t_{p-1}}{T}\right),$$(5)$$c_{N}=\frac{T}{2\pi^{2}n^{2}}\sum_{p=1}^{K}\frac{∆y_{p}}{∆t_{p}}\cos\left(\frac{2n\pi t_{p}}{T}\right)-\cos\left(\frac{2n\pi t_{p-1}}{T}\right),$$(6)$$d_{N}=\frac{T}{2\pi^{2}n^{2}}\sum_{p=1}^{K}\frac{∆y_{p}}{∆t_{p}}\sin\left(\frac{2n\pi t_{p}}{T}\right)-\sin\left(\frac{2n\pi t_{p-1}}{T}\right).$$

The statistical analysis of the coefficients for these equations revealed differences between sylvestris and cultivated seeds [30,34,35,36] and has been used in the study of fossil and archeological grape seeds [37,38,39,40,41,42,43]. The coefficients $a_{n},b_{n},c_{n},d_{n}$ of EFT equations can be extracted automatically with the software Momocs [44]. The number n of harmonics determines the degree of precision in the adjustment of the closed curve to the seed outline and it may vary depending on the objectives of the study. It was estimated that n=7 accounts for 95% of the total of shape variation in *Vitis* [37].

Fourier coefficients corresponding to the average outlines (Aos) of 271 cultivars of the Spanish collection at IMIDRA were obtained with Momocs [45], and the quantitative analysis of shape by *J*-index (percent similarity with models) followed by statistical analysis (PCA) revealed three groups: Group 1 containing all *Vitis* species other than *V. vinifera* and sylvestris seeds; Group 2 with seeds similar to Hebén; and Group 3 resembling the Chenin model [45]. The first objective of this study was to identify the seed images representing the varieties in the IMIDRA with the highest similarity to three models (Hebén, Chenin, and a new model based on Regina dei Vigneti and Muscat Hamburg). To achieve this, the *J*-index calculation procedure was modified to enable more precise outline adjustment by allowing changes in aspect ratio. This is based on the previous observation that changes in aspect ratio may be easier and consequently less significant than other morphological changes [45]. A second objective was to describe the differences between groups based on the geometric properties of their seed outlines. PCA enabled us to relate the Fourier coefficients to the geometric properties of the seeds, as well as to curvature measurements. This quantitative analysis of seed shape raises new hypotheses about the history of cultivars, which may contribute to the interpretation of molecular analysis results in representative genomes.

## 2. Results

### 2.1. A New Model Based on the Outlines of Regina Dei Vigneti and Muscat Hamburg

From their analysis of a large number of representative genomes, Dong et al. described Regina dei Vigneti and Muscat Hamburg as the most popular parental cultivars with a pure ancestry for one of the original genotypes in the basis of *V. vinifera* cultivars [25]. In consequence, a new model formed by the average silhouette of both cultivars was developed (Figure 1) to identify other cultivars associated with this group.

### 2.2. Three Groups Resulting from the Analysis of Shape Similarity (J-Index)

Three groups of cultivars were defined based on their outlines’ highest *J*-index values with the Hebén, Chenin, and Regina models: Group 1 comprises 15 cultivars, with the highest *J*-index values for the Hebén model (average outlines of Hebén 2020 and 2024 [10]), Group 2 comprises 25 cultivars, with the highest *J*-index values for the Chenin model [45], and Group 3 comprises 18 cultivars with the highest *J*-index values for the Regina model. The *J*-index values for the three models, as well as the values for solidity, aspect ratio (AR), circularity, curvatures, and 19 Fourier coefficients for the 58 selected cultivars and 6 EG varieties, are given in Zenodo (see Appendix A). Table 1 shows the comparison of means for *J*-index values, solidity, AR, roundness, and circularity. Differences in *J*-index, solidity, AR, roundness, and circularity were found between the three groups. Values of *J*-index are higher in the group corresponding to each model (Hebén with Hebén model, Chenin with Chenin model, and Regina with Regina model). Solidity is highest in the external group, followed by Hebén, and lowest in Chenin and Regina. Aspect ratio is higher in Regina, followed by Hebén and Chenin, and lowest in the external group. Roundness and circularity are highest in the external group, followed by Hebén and Chenin, and both values are lowest in Regina.

### 2.3. Cultivars in Each Group

The Hebén group (Figure 2) contains 15 cultivars, most of them in the progeny of Hebén (Merseguera, Mantúo de Pilas, Xarello, Planta Fina, Zalema, Jerónimo), or in the progeny of cultivars descending from Hebén (Juan García in the progeny of Cayetana Blanca, Tortosí from Beba). It also contains two cultivars of unkown origin (Tortozona Tinta, Terriza) and only two from crosses involving other cultivars (Planta Nova, Verdejo). It also includes the seeds of Hebén 2020 and 2024.

The Chenin group, with 25 cultivars (Figure 3), includes a smaller part of the progeny of Hebén (Cayetana Blanca, Mollar Cano, Derechero) as well as cultivars from many other crosses (Bastardo Negro, Tempranillo, Rojal Tinta, Torrontés, Moravía Dulce), a remarkable number of which are known to be in the progeny of Savagnin Blanc (Gewürztraminer, Bastardo Negro, Bruñal, Chenin, Prieto Picudo), or of unknown origin (Tinto Velasco, Ratiño, Riesling, Gualarido, Tinto de la Pampana Blanca, Garnacha, Vidadillo, Benedicto Falso, Albillo del Pozo). The change in aspect ratio allowed during the process of *J*-index calculation resulted in a better adscription of cultivars to this group (compare Figure 4 with Figure 5 of [45]).

The Regina group, with 18 cultivars (Figure 4), contains 8 of unknow origin (Bobal, Moravía Agria, Rayada Melonera, Bermejuela, Picapoll Tinto, Sabaté, Semillón, and De Cuerno), others in the progeny of Hebén (Corazón de Cabrito; Airén), Savagnin Blanc (Parduca), Castellana Blanca (Mazuela, Cariñena Blanca), and others from crosses involving exotic cultivars (Regina dei Vigneti, Muscat Hamburg, Malvasía Volcánica, Borba). Some of these are recent cultivars of table grapes (Red Globe).

### 2.4. PCA with the Fourier Coefficients Highlights Differences Between the Groups

PCA with the values of solidity, aspect ratio, circularity, roundness, and 19 Fourier coefficients (a2 to a7, b2 to b7, d1 to d7) showed differences between groups for all the binary combinations in Dim1 (PC1). In Dim2 (PC2), there were differences for all combinations except two (EG vs. Regina and Hebén vs. Chenin) (Figure 5). Figure 5 also shows the relation between the Fourier coefficients and geometric measurements. Aspect ratio (AR) and roundness (opposite to it) are related to Dim1, in association with coefficients A3, D1, D2, D3, D4, and D5. Dim2 grouped the coefficients A2, A4, B3, and B5.

Table A2 (Appendix B) contains the contributions of the measurements and coefficients to Dim1 and Dim2.

### 2.5. Relation Between the Fourier Coefficients and Curvature Measurements

The correspondence between coefficients and geometric properties becomes more complete when considering curvature values in the PCA. Table 2 presents the comparison of curvature values among the groups.

The distribution of groups considering curvature values in addition to Fourier coefficients is similar to the one in Figure 5, allowing to ascribe curvature values to some coefficients that remained unassociated to geometric properties (Figure 6). Thus, A2, A4, A7, B3, B5, B6, and D6 are associated with maximum curvature in the apical side and minimum curvature in the lateral side, composing a common factor. The other six curvature values are closely associated with roundness (and associated measurements) and constitute an orthogonal factor to the previous one.

### 2.6. Geometric Analysis of the Outlines in Representative Cultivars

Table 3 shows the cultivars corresponding to the points within a region defined by a 10% confidence interval or ellipse around each cluster’s centroid (Figure 5). The corresponding outlines are represented in Figure 7 (Hebén in upper line, A; Chenin in middle line, B; and Regina in bottom line, C), showing that solidity is highest in the Hebén group, and aspect ratio in Regina. In contrast to the compact, almost triangular form of the outlines in the Hebén group, a more prominent stalk is observed in the Chenin and Regina groups, being the cause of lower solidity in them. The stalk determines the region of maximum curvature at the apex (ApexMax), which is highest in Regina, as well as a region of negative curvature in the lateral side of the seeds (LatMin), whose absolute values are higher in the Chenin group. The difference between the Chenin and Regina groups lies in the shape of the stalk—more elongated and acute in Regina, rounded at the apex and with two parallel sides in Chenin—resulting in lower circularity and roundness in Regina than in the other two groups. In summary, while maximum apical curvature (ApexMax) is higher in Regina, minimum lateral curvature (LatMin) is higher (absolute value) in Chenin than in the other groups.

## 3. Discussion

The outline of the seed is treated here as a plane curve. Two important properties of closed-plane curves are aspect ratio and solidity. Aspect ratio (AR) is the inverse of roundness, and solidity is related to circularity, but independent of AR. Solidity gives an idea of compactness of a plane figure. Solidity is the most conserved property of the seed outlines in the groups of plants so far analyzed, including diverse species and families such as the Cucurbitaceae, Lamiaceae, and Euphorbiaceae [46,47,48,49,50], as well as in species and cultivars of *Vitis* [29,45] (Table 1). In the classical studies of *Silene* L. (Caryophyllaceae) by Boissier, solidity was the basis for the differentiation between two subgenera based on the shape of their seeds: *dorso plana* (high solidity) and *dorso canaliculata* (low solidity) [51], formed today by subgenera *Behenantha* and *Silene*, respectively [52].

The seeds of wild-grown *Vitis* species tend to have high solidity values [29], and during the history of *Vitis* cultivation a general trend towards a reduction in solidity has been associated with increased AR. The predominance of rounded forms in the seeds of wild *Vitis* species including *V. vinifera* subsp. *sylvestris* versus elongated- pyriform seeds in many cultivars has been noticed by many authors [30,34,35,36,37,38,39,40,41,42,53,54,55,56,57,58]. Increased nutrient availability during cultivation resulted in larger berries with elongated seeds of higher AR and lower solidity than their wild counterparts. In recent cultivars, the outline became more sinuous with changes in the direction of curvature. Thus, the seeds of the group formed by Hebén and related cultivars resemble more those of *Vitis vinifera* subsp. *sylvestris* and have lower AR and higher solidity than modern cultivars (Figure 2, Figure 5, and Figure 7), while recent cultivars like the table variety Red Globe are characterized by elongated seeds with long stalks, associated with high AR and low solidity values. A gradient in solidity goes from *Vitis* species other than *V. vinifera* and sylvestris genotypes to the group Hebén, in an intermediate position, and to modern cultivars of the Chenin and Regina dei Vigneti [28,29,45] groups. Differences between the two later groups consist of AR, lower in Chenin and higher in Regina, concomitantly with higher roundness and circularity values, and differences in curvature.

In the representations of Elliptic Fourier Transform, the geometric properties of plane curves are encoded in the Fourier coefficients. In the PCA reported here, the differences between groups of cultivars were related with the main geometric properties both in PC1 and PC2, being AR inverse to roundness and determining between these two properties a trend close to PC1 (Table 2). Both circularity and solidity are closely aligned with PC1 (Table 2). These measurements were related to D1, D2, D3, D4, D5 and A3. In general, the coefficients for the lower harmonics determine basic aspects of the geometry of the curves such as AR and solidity, while those of the higher harmonics are responsible for more subtle changes in the curve, with maximum values of curvature in the apex and minimum values in the lateral side of the seeds associated with coefficients A2, A4, A7, B3, B5, B6 and D6.

The seeds of the Hebén group represent an intermediate status in the development of cultivars, with lower values of solidity than sylvestris seeds, but higher than most recent varieties. Some cultivars in the offspring of Hebén have seed outlines with high values of AR and low solidity corresponding to the Chenin morphotypes. This is the case of Mollar Cano, whose seeds harvested in two years have the straight stalk characteristic of the Chenin type. It is possible to verify whether this morphological difference between Hebén and Mollar Cano is attributable to a single genetic factor, and this may be achieved by crosses. If this is the case, then Mollar Cano may have inherited the shape of their seeds from an unknown progenitor, most probably related to the Chenin group, but it seems not very plausible that this characteristic may come from a *Vitis vinifera* subsp. *sylvestris* progenitor. On view of these results, it is tempting to speculate that the main geometric properties of the *Vitis* seed may be a tool to determine the status of the seeds and to differentiate between seeds from cultivars (or ferals) and sylvestris.

The seed morphotype Chenin occurs in cultivars of the family known as Traminer, related to Savagnin, an ancient and highly influential group with variations rich in color and aroma such as those of Savagnin Blanc, Gewürztraminer, and Savagnin Rosé, due to somatic mutations accumulated over centuries of vegetative propagation. It is related with Cabernet, Syrah and Merlot, and the finding of the Spanish variety Tempranillo associated to this large an important group agrees with the results of Terral et al. [30]. Thus, the groups resulting in this work based on seed shape agree with other studies based on seed morphology [30], as well as data from pedigree analysis based on nuclear microsatellite markers [9]. Further *J*-index and curvature analysis with other cultivars may help to investigate differences in seed morphology between those groups.

The variety Königin der Weingärten (Regina dei Vigneti) was developed in Hungary by Mathiasz Janos in 1916 by crossing Afus Ali and Csaba Gyoenye [59,60]. The variety Muscat Hamburg was obtained in the United Kingdom by Seward Snow, crossing Schiava Grossa and Muscat of Alexandria [61,62]. The ADMIXTURE analysis in the work of Dong et al. [25] using these cultivars obtained through genetic improvement may skew genetic and varietal variability studies attributing a similar statistical value to genomic regions with adaptative value in some ancient varieties and others that are transient, as the result of recent crosses.

The morphological differences between the seeds of *Vitis* species and *V. vinifera* subsp. *sylvestris* on one side and *V. vinifera* subsp. *vinifera* cultivars on the other side have been reported [29,30,31]. The results presented here support those findings and point towards a method to differentiate between *sylvestris* and feral grapes. The results of genomic analysis suggesting introgression with local *sylvestris* plants must be re-evaluated considering that in some instances these may be feral instead of real *sylvestris* genotypes.

## 4. Materials and Methods

### 4.1. Plant Material

This works departs from a selection of cultivars based on the similarity of their seed outlines to geometric models measured by the *J*-index. From the 271 *Vitis* species and cultivars described in [45], 58 were selected based on the similarity of the outlines based on their EFT coefficients with three geometric models: 15 cultivars with high similarity to the Hebén model (average seed shape of Hebén seeds in 2020 and 2024), 25 cultivars of high similarity to the Chenin model (average of Chenin and Gewürztraminer) and 18 cultivars of high similarity to the Regina model (average of Regina dei Vigneti and Muscat Hamburg). In addition to these 58 cultivars, an external group (EG) was formed by three *Vitis* species different from *V. vinifera* and three sylvestris genotypes. The “sylvestris” samples were female plants surveyed between 2003 and 2009 in natural populations of riparian forests in the provinces of Cádiz and Navarra (Spain). Sylvestris plants can either belong to *V. vinifera* subsp. *sylvestris* or be ferals (plants from cultivars that have escaped cultivation). Most of the seeds used in this study were harvested in 2020 as described by Espinosa-Roldán et al. [36]. Cultivars harvested in 2024 were described by Cervantes et al. [10]. The list of 64 cultivars is given in Table A1 in Appendix B.

### 4.2. Photography

Images of the seeds collected in August 2020 were taken at IMIDRA (Madrid), while images of the remaining seeds were taken at IRNASA (Salamanca). In both cases, an F-stop of f/18 was used, with lighting of between 800 and 1200 lumens and a color temperature of 6400 K. No filters were used to clean up the images. In Madrid, a Nikon D80 10.2-megapixel camera (Nikon, Tokyo, Japan), equipped with a COSINA 100 mm f/3.5 MC Macro AF (Cosina Co., Ltd., Nakano, Japan) with an ISO of 400 was used, and a Sony α5100 24-megapixel camera (Sony, Tokyo, Japan) with an AF-S Micro NIKKOR 60 mm f/2.8G ED lens (Nikon, Tokyo, Japan) with an ISO of 100 and an object distance of 17 cm was used in Salamanca. The shutter speed was set manually to 1/5 sec and the aperture to F/16. In both cases, the ISO values were set manually. The focal distance of the objectives guarantees minimal distortion. Additionally, the photographs of the seeds occupied the central part of the image, leaving the edges unoccupied to avoid distortion. Color correction is not required as the images are transformed into corresponding black contours. The image quality is sufficient (ISO values of 400 or less guarantee that noise-reducing digital filters are not required). The original seed images are available at Zenodo (see Appendix A).

### 4.3. General Morphological Measures

Measurements of the seed images (area, perimeter, length, width, circularity, aspect ratio, roundness, and solidity) were taken in 20 seeds for cultivar using ImageJ 1.54h [63]. The program converts pixels into millimeters according to a ruler contained within the images. To do this, the cursor is placed at the start of the ruler and a line corresponding to the ruler is drawn using the “Analyse”, “Set Scale” function. For example, the average area of 30 Hebén seeds (2020) is 42,320 pixels (21 mm^2^). After adjusting the pixel-to-millimeter ratio, the images were converted to 8-bit and the threshold adjusted before taking the measurements using the “Analyse particles” function. Circularity, aspect ratio, roundness, and solidity are described in detail elsewhere [60,61,62]. Circularity is the ratio:(7)$$C=\frac{4\;\pi \;A}{P^{2}}$$
where A is area and P is perimeter. Aspect ratio is the quotient L/W, where L is the length and W the width. Roundness is determined by:(8)$$R=\frac{4\;A}{\pi \;L^{2}}.$$

Solidity is a property of closed-plane curves related to their convexity. It expresses the ratio of the area of an object to the area of its convex hull (the convex hull is the smallest convex set that contains a plane figure). Solidity is here given ×1000.

### 4.4. Extraction of Fourier Coefficients from the Images with Momocs

Images containing 20 or 30 seeds were converted into TPS files, with the coordinates for each seed outline extracted in accordance with the protocol outlined in Momocs [44]. The corresponding EFT coefficients were then applied to Mathematica equations to determine the average outline relative to each cultivar. The Aos for all cultivars and species were described by Martín Gómez et al. [45]. According to bilateral symmetry of the images, the c coefficients c1 to c7 were shown not to contribute significantly to the equation and were discarded, leaving only a1 to a7, b1 to b7, and d1 to d7 for the analysis (a1 = 1; and b1 = 0). In consequence, the parametric equations representing the curves had 19 variable coefficients belonging to n=7 harmonics.

### 4.5. Models and J-Index Measurements

*J*-Index gives the percent similarity between a seed outline and a given model. The Hebén and Chenin models were as described [45], while the Regina model was designed for this work as the average outline of Muscat Hamburg and Regina dei Vigneti, two cultivars considered to represent the Muscat group, one of the six precursor groups based on genome analysis [25].

The *J*-index is calculated by comparing two images: a seed outline corresponding to the average of a variety and the model. For this, the models are superimposed on images of outlines of each cultivar, looking for maximum similarity. The models are overlaid on Corel PhotoPaint 24.5.0.731 (Corel Corporation; Ottawa, ON, Canada) containing outlines of each cultivar, and two new files are saved for each of these images: one with the model in black and one with the model in white. The ImageJ program [63] gives the values of total area (T, being the contour of the model in black, the whole area is considered) and area shared between the model and the seeds (S, being the contour of the model in white, where the measured area is limited to the area shared between the seed and the model; Figure 8). Note that “T” is the total surface occupied by either the seed or the model, i.e., the number of pixels, whereas in “S” the measured surface is shortened by the white line corresponding to the profile of the model. For each seed, the *J*-index is calculated as the ratio S/T. Unlike previous measurements, this work allowed for a change in aspect ratio during adjustment. The images used for the *J*-index calculation are available at Zenodo (see Appendix A).

Three groups were formed with those cultivars having the highest *J*-index with each of the models. Unlike the previous study, in which groups were formed with all varieties that yielded higher *J*-index values with each model, in this case a condition was added to form the groups: it was not enough for the *J*-index values to be higher with each model, but the difference with the other two models had to be equal to or greater than one. A fourth group made by three *Vitis* species different from *V. vinifera*, and three sylvestris cultivars, of the highest solidity, formed the external group of wild grapes (EG) in the PCA.

### 4.6. Curvature Analysis

Curvature analysis of seed silhouettes was conducted using images with a resolution of 150 ppi, and the maximum, minimum, and average curvatures were determined for each seed’s lateral view. A series of points delineating the seeds’ profiles was automatically obtained using the Analyze Line Graph function in ImageJ, and the corresponding Bézier curve and curvature measurements were obtained according to published protocols [64]. Curvature is provided in mm^−1^; thus, a curvature of 1 corresponds to a circumference of 1 mm and a curvature of 10 to a circumference of 100 µm (0.1 mm). Maximum and minimum curvature values indicate major changes in slope of the curve. Curvature equals 0 in a straight line and is constant in a circumference. Therefore, the greater the difference between the maximum and minimum curvature values and the average curvature, the greater the departure from circularity. A low maximum-to-average curvature ratio indicates more constant curvature values, which are associated with figures approaching a circumference. Negative values indicate changes in slope direction and thus non-convex regions. Curvature values included maximum, minimum, average, and maximum-to-average curvature ratio in the apical and lateral regions of the seed outline, as indicated in Figure 9. Representative Mathematica files for curvature analysis are available on Zenodo (see Appendix A).

### 4.7. Statistical Analysis

As some of the populations did not follow a normal distribution, non-parametric tests were applied for the comparison of populations. Kruskal–Wallis tests were conducted followed by stepwise stepdown comparisons by the ad hoc procedure developed by Campbell and Skillings. Differences between populations were indicated in the tables by different superscript letters. Statistical analysis was performed with IBM SPSS statistics v29 (SPSS 2022). PCA was conducted with R [65].

## 5. Conclusions

Based on values of *J*-index (percent similarity with geometric models) three groups of *Vitis* seeds were defined corresponding to Hebén, Chenin, and Regina dei Vigneti. The morphological and geometrical analysis showed differences between the groups, as well as differences between each of the morphotypes and the external group (EG) of seeds from diverse species of *Vitis* and sylvestris plants. PCA revealed an association between geometric properties (circularity, curvature values, roundness, solidity) and Fourier coefficients. Departing from accurate models, it is possible to investigate family relationships between *Vitis* cultivars.

## Figures and Tables

**Figure 1 plants-14-02481-f001:**
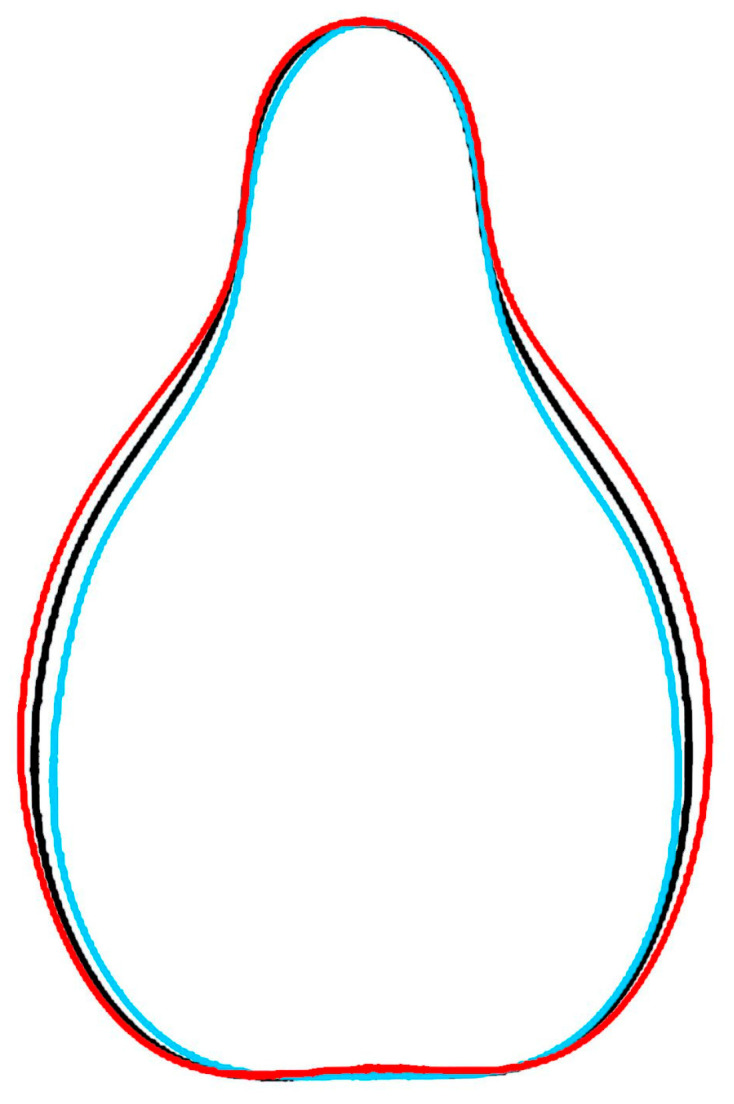
The Regina model (black) is the average outline of the seeds of Regina dei Vigneti (red) and Muscat Hamburg (light blue).

**Figure 2 plants-14-02481-f002:**
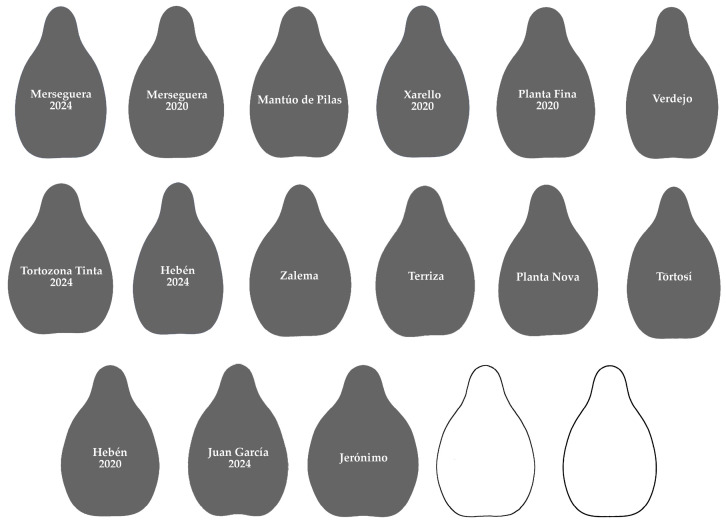
Aos corresponding to Hebén group. Individual images are ordered by decreasing similarity with the Hebén model (from highest to lowest values of *J*-index). The last two images correspond to average outline of 15 seeds of the Hebén group and model, respectively.

**Figure 3 plants-14-02481-f003:**
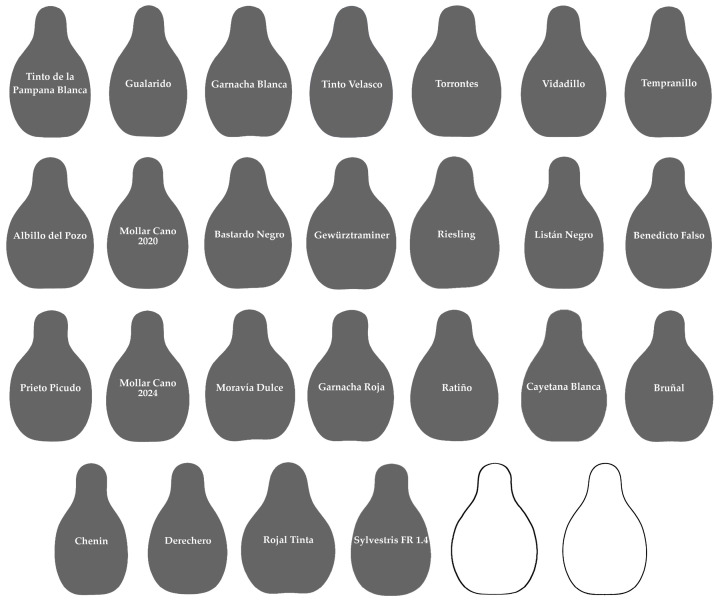
Group of 25 cultivars that gave the highest values of *J*-index with the Chenin model. During the process of *J*-index calculation it was allowed to change aspect ratio, resulting in a better adscription of cultivars to this group. Individual images are ordered by decreasing similarity with the Chenin model (from highest to lowest values of *J*-index). The last two images correspond to the average outline of 15 seeds of the Chenin group and model, respectively.

**Figure 4 plants-14-02481-f004:**
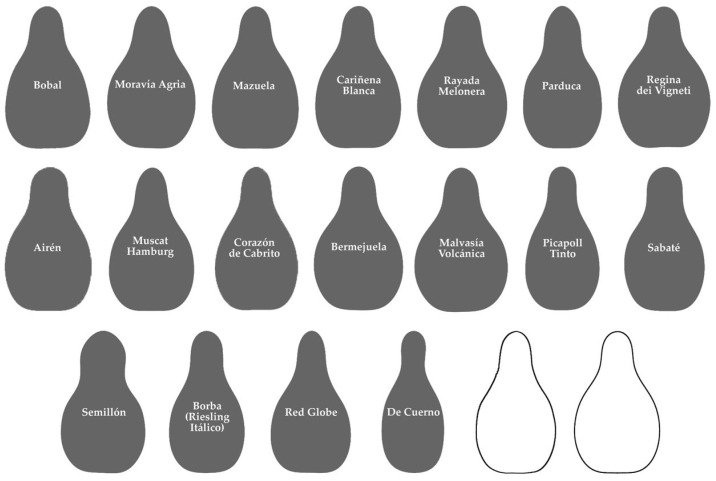
Group of 18 cultivars that gave the highest values of *J*-index with the Regina model. Individual images are ordered by decreasing similarity with the Regina model (from highest to lowest values of *J*-index). The last two images correspond to average outline of 15 seeds of the Regina group and model, respectively.

**Figure 5 plants-14-02481-f005:**
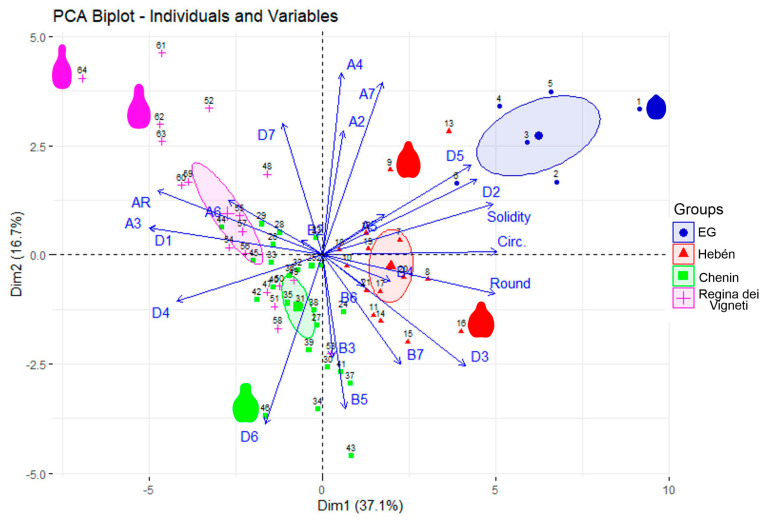
PCA biplot representing the cultivars in the groups and the relative influence of the morphological measurements on the distribution. Numbers of the cultivars correspond with those given in Table A1 in Appendix B. Outlines of representative cultivars are shown: 1, *Vitis amurensis*; 9, Juan García 2024; 16, Jerónimo; 46, Garnacha Roja; 62, Borba (Riesling Itálico); 64, De Cuerno. These are represented in the respective positions in Figure 2, Figure 3 and Figure 4 (for example, de Cuerno at the end of Figure 4). EG: external group.

**Figure 6 plants-14-02481-f006:**
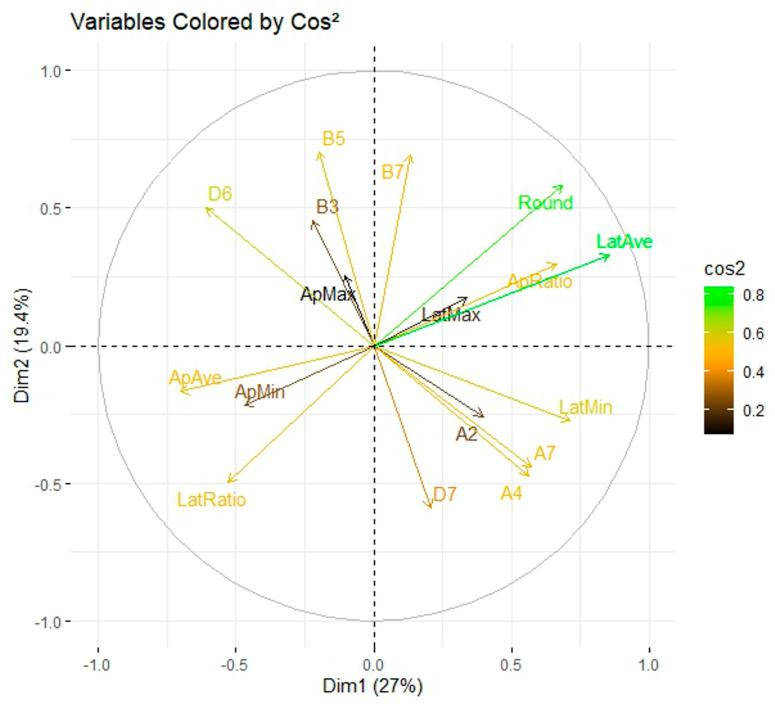
Correlation circle plot or variable factor map. In addition to roundness, curvature measurements appear to be related to the coefficients. The max-to-average curvature ratio (lateral) is opposed to roundness, while the max-to-average curvature ratio (apical) is closer to solidity. Average curvature (apical) is opposed to curvature ratio (apical). Maximum apical curvature is positively associated with B3, B5, B7, and D6 and negatively with D7. Minimum lateral curvature is positively associated with A2, A4, and A7 and negatively with D6.

**Figure 7 plants-14-02481-f007:**
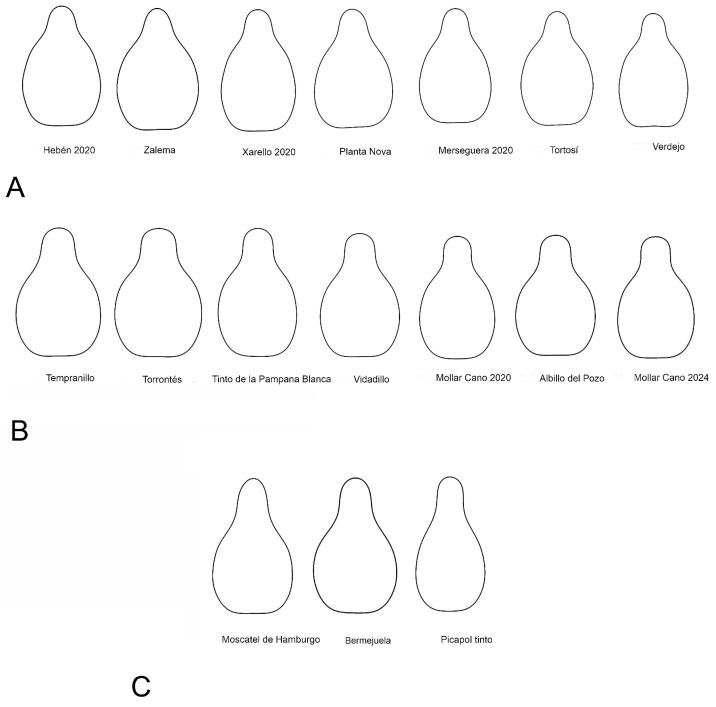
Seed outlines of cultivars in the 10% confidence ellipses of three morphological groups. (**A**) Hebén; (**B**) Chenin; (**C**) Regina.

**Figure 8 plants-14-02481-f008:**
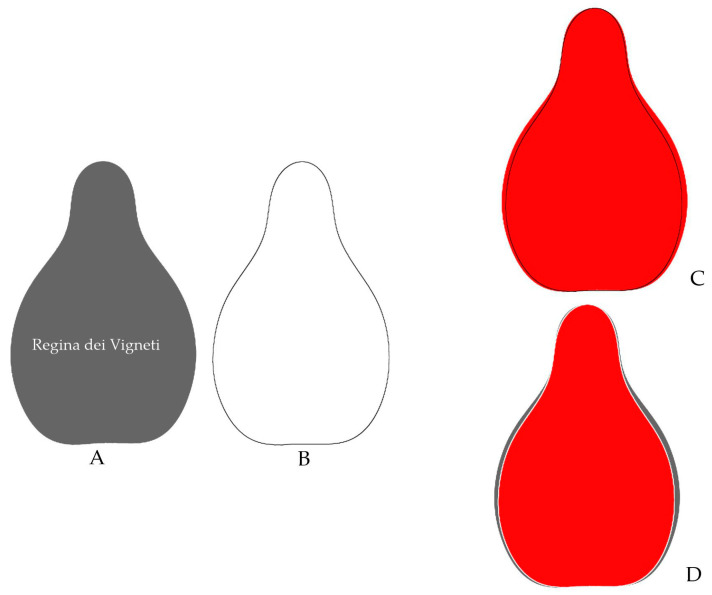
Schematic representation of the method to calculate *J*-index (percent of similarity of a set of images with a model). The model (average silhouette of Hebén, (**B**)) is superimposed here to an outline of Regina dei Vigneti (**A**), searching the maximum surface shared between the image and the model. In the right, above, the superimposed model is in black (**C**), and below, in white (**D**). When the images on the left are opened in ImageJ (right, red), the image on top (**C**) shows the total number of pixels in the figure (corresponding to the surface occupied by the red color), while the image below (**D**), shortened by the white line corresponding to the profile of the model, shows the shared surface (also in red). The number of pixels shared by the image and the model is lower or equal to the total.

**Figure 9 plants-14-02481-f009:**
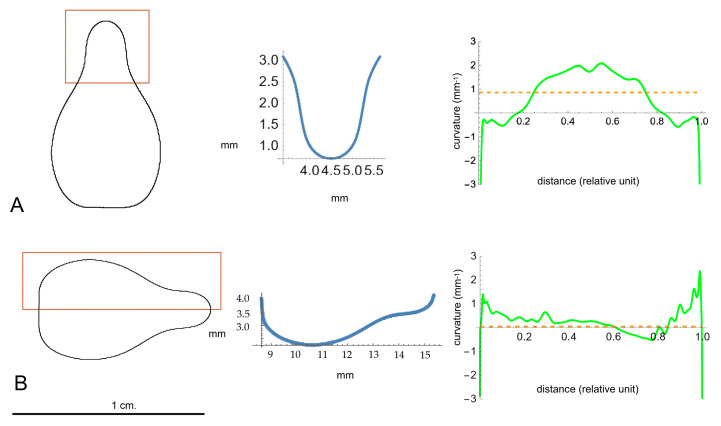
Method for calculation of curvature. Above (**A**): curvature in the apex. Below (**B**): lateral curvature. Left: average outline of seed (Corazon de Cabrito) indicating region to measure curvature. Center: Bézier curve corresponding to fragment of outline. Right: curvature values along the Bézier curve. Curvature was evaluated between 0.2 and 0.8 to avoid conflicting regions in extremes of the Bézier curve.

**Table 1 plants-14-02481-t001:** Results of Kruskal–Wallis comparison between mean values for *J*-index with the Hebén, Chenin, and Regina models, solidity, aspect ratio (AR), roundness, and circularity in the groups of cultivars. Between brackets: coefficients of variation. Different letters in superscript in consecutive rows in each column indicate significant differences. EG: external group formed by *Vitis* species other than *V. vinifera* and sylvestris grapes. N = number of cultivars in each group.

Group	N	*J*-Index Hebén	*J*-Index Chenin	*J*-Index Regina	Solidity	AR	Roundness	Circularity
EG	6	87.0 ^a^ (3.3)	87.8 ^a^ (2.4)	84.3 ^a^ (2.3)	994 ^c^ (0.1)	1.35 ^a^ (8.2)	0.74 ^c^ (7.7)	0.83 ^c^ (3.1)
Hebén	15	96.6 ^d^ (0.5)	95.1 ^c^ (0.5)	93.1 ^b^ (0.8)	975 ^b^ (0.3)	1.56 ^b^ (5.7)	0.64 ^b^ (5.8)	0.74 ^b^ (2.2)
Chenin	25	95.1 ^c^ (0.5)	96.5 ^d^ (0.3)	93.5 ^b^ (1.0)	958 ^a^ (0.6)	1.61 ^b^ (6.0)	0.62 ^b^ (5.9)	0.73 ^b^ (2.3)
Regina	18	92.8 ^b^ (1.5)	92.9 ^b^ (1.4)	95.6 ^c^ (1.4)	953 ^a^ (0.9)	1.82 ^c^ (10.9)	0.56 ^a^ (9.7)	0.68 ^a^ (4.8)

**Table 2 plants-14-02481-t002:** Results of Kruskal–Wallis comparison between curvature values in morphological groups. Ap = apical; Lat = lateral. Maximum (Max), minimum (Min), average (Ave), and max-to-average ratio (Ratio) curvatures. Different letters in superscript in consecutive rows in each column indicate significant differences. EG: external group formed by *Vitis* species other than *V. vinifera* and sylvestris grapes. N = number of cultivars in each group.

Group	N	ApMax	ApMin	ApAve	ApRatio	LatMax	LatMin	LatAve	LatRatio
EG	6	1.91 ^ab^ (21.9)	0.26 ^a^ (22.8)	0.50 ^a^ (13.1)	3.84 ^b^ (23.6)	0.59 ^b^ (5.8)	−0.19 ^c^ (60.2)	0.19 ^d^ (17.9)	3.12 ^a^ (15.7)
Hebén	15	1.92 ^ab^ (16.6)	0.37 ^a^ (83.3)	0.83 ^b^ (11.2)	2.30 ^a^ (13.0)	0.45 ^a^ (14.9)	−0.63 ^b^ (20.6)	0.10 ^c^ (20.9)	4.85 ^b^ (21.5)
Chenin	25	1.90 ^a^ (22.7)	0.85 ^b^ (31.8)	0.80 ^b^ (12.8)	2.39 ^a^ (21.2)	0.50 ^a^ (18.1)	−0.74 ^a^ (15.9)	0.08 ^b^ (30.4)	6.61 ^c^ (29.6)
Regina	18	2.06 ^b^ (14.2)	0.77 ^ab^ (69.3)	0.88 ^b^ (24.7)	2.43 ^a^ (27.4)	0.49 ^a^ (18.7)	−0.55 ^b^ (21.5)	0.05 ^a^ (26.6)	9.66 ^d^ (24.1)

**Table 3 plants-14-02481-t003:** Cultivars in the centroid of each group (10% confidence level). Numbers of the cultivars correspond with those given in Table A1 in Appendix B.

Hebén	Chenin	Regina
7. Hebén 2020	27. Tempranillo	55. Muscat Hamburg
8. Zalema	31. Torrontés	57. Bermejuela
12. Xarello 2020	32. Tinto de la Pampana Blanca	59. Picapoll Tinto
17. Planta Nova	35. Vidadillo	
19. Merseguera 2020	36. Mollar Cano 2020	
20. Tortosí.	39. Albillo del Pozo	
21. Verdejo	40. Mollar Cano 2024	

## Data Availability

Research data published in the article and Appendix A.

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
