# Peer review of "Morphometric Analysis Reveals New Data in the History of Vitis Cultivars"

_plants, 2025, doi:10.3390/plants14162481_

Round 1

Reviewer 1 Report

Comments and Suggestions for Authors

The experimental design is generally reasonable but need to mention how many seeds each cultivar or subspecies are used for taking images. The manuscript is generally well prepared but many places are not inconistent in spelling style or form. Pls read carefully and make corrections as needed, I just marked some places as examples for reference.

The morphology of seeds may provide a clue of history but may not be robust in evidence since the seeds may be diverse in shapes and size, particularly gene flow happened in difference populations due to a long history of cultivation. If the molecular data are included to generate a phylogeny tree, it will be more informative.

Author Response

Dear reviewer,

Thank you very much for your commentaries and corrections that have contributed to improve the quality of the article. 

Q1: The experimental design is generally reasonable but need to mention how many seeds each cultivar or subspecies are used for taking images. The manuscript is generally well prepared but many places are not inconistent in spelling style or form. Pls read carefully and make corrections as needed, I just marked some places as examples for reference.

A1:The manuscript has been modified following your recommendations. All the corrections indicated in the PDF have been made.

Q2: The morphology of seeds may provide a clue of history but may not be robust in evidence since the seeds may be diverse in shapes and size, particularly gene flow happened in difference populations due to a long history of cultivation. If the molecular data are included to generate a phylogeny tree, it will be more informative

This study is based on seed morphometric analysis.  Unfortunately, we do not have the original molecular data required to construct a phylogenetic tree as requested. Nevertheless, the agreement between the data here reported and other studies including molecular data has been expressed now in the Discussion section:

The groups resulting in this work based on seed shape agree with other studies based on seed morphology [30], as well as data from pedigree analysis based on nuclear microsatellite markers [9]. Further J-index and curvature analysis with other cultivars may help to investigate differences in seed morphology between those groups.

Reviewer 2 Report

Comments and Suggestions for Authors

The manuscript by Martín-Gómez et al. aims to digitally describe grape seeds from various cultivars using descriptors based on the Fourier transform and additional seed geometrical characteristics (solidity, roundness, aspect ratio, etc.). This work builds upon previous research (ref 45) by introducing a new cultivar group (Regina) and analyzing outline curvature characteristics.

General Observations:

Overall, the paper is well-written, and the results appear interesting. However, further clarification is needed in the results and methods sections.

Specific Recommendations:

1) Indices (i, p, n): Provide descriptions for indices (i, p, n) in lines 79-92.

2) Table 1: There is no clear correspondence between superscript letters, significant differences, and compared groups. More detailed descriptions are necessary. Additionally, rewrite lines 135-139, as they are unclear. The same applies to Table 3.

3) Line 140 & Table 3: The description for "Kruskal-Wallis comparison between mean values" was not provided in the Methods section. What hypothesis was tested? The Mann–Whitney U test is mentioned in the 4.7 Statistical Analysis section (lines 445-448). Is this correct? Clarify the statistical method used for Kruskal-Wallis comparisons.

4) Figure 2: It is challenging to compare multiple seed images visually. Is an order to the images is similar to that from ref 45? If so, the figure caption should provide this information.

If the goal is to describe variations between seeds, consider superimposing all seeds and highlighting variable parts of the outline. Alternatively, show variability in shape parameters (e.g., boxplots for normalized or comparable values) to determine which parameters are most variable.

In the Discussion section (lines 254-256), the authors mention, "Solidity is the most conserved property of the seed outlines in the groups of plants so far analyzed, including diverse species and families." Can this conservation be observed for grape seeds in this study?

5) Similar recommendations apply to Figures 3 and 4. Consider characterizing each cultivar group using these figures.

6) Figure 5: The names of the groups (Regina, Chenin, Hebén, EG) should be added directly in the figure.

The numbers near the dots represent should be clarified. If these are cultivar numbers, please indicate this in the figure and Table 4 for consistency.

I recommend adding outlines for characteristic cultivars in the plot (e.g., leftmost, rightmost, topmost, or representatives from the groups) to improve understanding of shape variability.

7) Figure 5: Cultivar #64 appears to be an outlier for group 4. I wonder, is it represented in  Figs 2, 3, or 4 for context.

8) Figure 5 & Figure 6: I recommend to represent Figure 5 as a biplot instead of having a separate Figure 6 for variable correlations. This could help interpret the directions of variability in the PCA plot.

I am not sure that Figures 5 and 6 have PC in the same direction, since the values of the explained variation differ.

9) Table 2: Is this table necessary in the main text? The information in Fig 6 and Table 2 seems redundant. Consider removing one.

10) Figures 6 and 7 Captions: I recommend to move any remarks about plot interpretation from the figure captions to the main text for better flow.

11) Table 4: Clarify the meaning of the numbers next to the cultivar names.

12) Figure 8: I recommend to add panels related to different groups and name them in the caption. The current format makes it difficult to find the correspondence between the figure and its description (lines 235-246).

13) Lines 268-272 (Discussion Section): "Seeds of the group formed by Hebén and related cultivars resemble more those of Vitis vinifera subsp. sylvestris and have lower aspect ratio and higher solidity than modern cultivars." Please refer to tables/figures in the Results section to support this conclusion.

14) Line 401: "Note that 'T' is the total surface occupied by either the seed or the model." Clarify what is meant by "surface" in this context.

15) Figure 9: I recommend to add notation for the T and S areas in the figure (use color or arrows to mark them).

16) Please provide panel notation (a, b, c, d) and refer to them in the figure caption.

17) Lines 416-417: "Those cultivars with a difference in the J-index value obtained with two models lower than 1 were discarded." This sentence is unclear. Please rephrase for clarity.

18) Figure 10: Separate images into panels in the figure and provide references to them in the figure caption. Sign axes in the middle and right images.

Author Response

Comments 1: 

The manuscript by Martín-Gómez et al. aims to digitally describe grape seeds from various cultivars using descriptors based on the Fourier transform and additional seed geometrical characteristics (solidity, roundness, aspect ratio, etc.). This work builds upon previous research (ref 45) by introducing a new cultivar group (Regina) and analyzing outline curvature characteristics.

General Observations:

Overall, the paper is well-written, and the results appear interesting. However, further clarification is needed in the results and methods sections.

Response 1:

Dear reviewer,

Thank you very much for your commentaries and corrections that have contributed to improve the quality of the article. The manuscript has been modified following your recommendations. All the corrections indicated in the PDF have been made.

Comments 2: Specific Recommendations:

Q1) Indices (i, p, n): Provide descriptions for indices (i, p, n) in lines 79-92.

A1) Descriptions are given now in lines 86-91.

Q2) Table 1: There is no clear correspondence between superscript letters, significant differences, and compared groups. More detailed descriptions are necessary. Additionally, rewrite lines 135-139, as they are unclear. The same applies to Table 3.

A2) Lines 135-139 (159-164 now) have been rewritten for more clarity.

Q3) Line 140 & Table 3: The description for "Kruskal-Wallis comparison between mean values" was not provided in the Methods section. What hypothesis was tested? The Mann–Whitney U test is mentioned in the 4.7 Statistical Analysis section (lines 445-448). Is this correct? Clarify the statistical method used for Kruskal-Wallis comparisons.

A3) The corresponding sentence (Section 4.7. Statistical analysis) has been corrected to:

As some of the populations did not follow a normal distribution, non-parametric tests were applied for the comparison of populations. Kruskal–Wallis tests were done followed by stepwise stepdown comparisons by the ad hoc procedure developed by Campbell and Skillings. Differences between populations were indicated in the tables by different superscript letters.

Q4-1) Figure 2: It is challenging to compare multiple seed images visually. Is an order to the images is similar to that from ref 45? If so, the figure caption should provide this information.

A4-1) Individual outline images are ordered by decreasing similarity with the model from highest to lowest values of J-index. This has been indicated now in the corresponding figure legends (Figures 2, 3 and 4), for example:

Individual images are ordered by decreasing similarity with the model Regina (from highest to lowest values of J-index).

Q4-2) If the goal is to describe variations between seeds, consider superimposing all seeds and highlighting variable parts of the outline. Alternatively, show variability in shape parameters (e.g., boxplots for normalized or comparable values) to determine which parameters are most variable.

A4-2) Figures of the average outlines of all the contours represented and the corresponding models have been added to Figures 2, 3 and 4. This is now indicated in the figure legends, for example:

The last two images correspond to average outline of 15 seeds of the group and model Regina, respectively.

Q4-3) In the Discussion section (lines 254-256), the authors mention, "Solidity is the most conserved property of the seed outlines in the groups of plants so far analyzed, including diverse species and families." Can this conservation be observed for grape seeds in this study?

A4-3) Related to this aspect, the following information has been added to this sentence:

Solidity is the most conserved property of the seed outlines in the groups of plants so far analyzed including diverse species and families such as the Cucurbitaceae, Lamiaceae and Euphorbiaceae [46-50], as well as in species and cultivars of Vitis [29,45] (Table 1).

Q5) Similar recommendations apply to Figures 3 and 4. Consider characterizing each cultivar group using these figures.

A5) Average outlines as well as the outline of the corresponding model have been added to Figures 2, 3 and 4 to characterize each group. The following or a similar sentence has been added to the figure legends:

The last two oulines correspond to average outline of 15 seeds of the group and model Hebén, respectively.

Q6) Figure 5: The names of the groups (Regina, Chenin, Hebén, EG) should be added directly in the figure.

The numbers near the dots represent should be clarified. If these are cultivar numbers, please indicate this in the figure and Table 4 for consistency.

I recommend adding outlines for characteristic cultivars in the plot (e.g., leftmost, rightmost, topmost, or representatives from the groups) to improve understanding of shape variability.

A6) The names of the groups (Regina, Chenin, Hebén, EG) have been added directly in the figure. The following sentence has been added to the legend of Figure 5:  Numbers of the cultivars correspond with those given in Table A1 in Appendix A.

Outlines have been added for the most characteristic cultivars in the plot.

Q7) Figure 5: Cultivar #64 appears to be an outlier for group 4. I wonder, is it represented in  Figs 2, 3, or 4 for context.

A7) Cultivar #64 is De Cuerno. The outline has a particularly high AR value and has been represented now in Figure 5. The legend to Figure 5 indicates also that this outline is represented in Figure 4.

Q8) Figure 5 & Figure 6: I recommend to represent Figure 5 as a biplot instead of having a separate Figure 6 for variable correlations. This could help interpret the directions of variability in the PCA plot.

I am not sure that Figures 5 and 6 have PC in the same direction, since the values of the explained variation differ.

A8) Figure 5 has been changed to a Biplot as suggested. Figures 5 and 6 of the previous version have been merged in the actual Figure 5.

Q9) Table 2: Is this table necessary in the main text? The information in Fig 6 and Table 2 seems redundant. Consider removing one.

A9) Table 2 has been moved to Appendix A and is now Table A2.

Q10) Figures 6 and 7 Captions: I recommend to move any remarks about plot interpretation from the figure captions to the main text for better flow.

A10) Plot interpretations have been removed from the figure legends to the main text.

Q11) Table 4: Clarify the meaning of the numbers next to the cultivar names.

A11) Similar to Figure 5, the legend to Table 4 indicates now the meaning of the numbers:

Numbers of the cultivars correspond with those given in Table A1 in Appendix A.

Q12) Figure 8: I recommend to add panels related to different groups and name them in the caption. The current format makes it difficult to find the correspondence between the figure and its description (lines 235-246).

A12) Panels corresponding to the three groups have been added (A: Hebén; B: Chenin; C: Regina), and this has been indicated in the main text and in the Figure legend.

Q13) Lines 268-272 (Discussion Section): "Seeds of the group formed by Hebén and related cultivars resemble more those of Vitis vinifera subsp. sylvestris and have lower aspect ratio and higher solidity than modern cultivars." Please refer to tables/figures in the Results section to support this conclusion.

A13) Figures 2, 5 and 7 are now quoted in this sentence.

Q14) Line 401: "Note that 'T' is the total surface occupied by either the seed or the model." Clarify what is meant by "surface" in this context.

A14) Surface is now explained in the text as the number of pixels:

Note that “T” is the total surface occupied by either the seed or the model, i.e. the number of pixels, whereas in “S” the measured surface is shortened by the white line corresponding to the profile of the model.

Q15) Figure 9: I recommend to add notation for the T and S areas in the figure (use color or arrows to mark them).

A15) surface is explained as the number of pixels. Both T and S areas are in Red, and the difference is explained in the Figure legend:

When the images on the left are opened in Image J (right, red), the image on top (C) shows the total number of pixels in the figure (corresponding to the surface occupied by the red color), while the image below (D), shortened by the white line corresponding to the profile of the model, shows the shared surface (also in red). The number of pixels shared by the image and the model is lower or equal to the total.

Q16) Please provide panel notation (a, b, c, d) and refer to them in the figure caption.

A16) Panel notations have been added (A, B, C, D) and are now indicated in the figure caption.

Q17) Lines 416-417: "Those cultivars with a difference in the J-index value obtained with two models lower than 1 were discarded." This sentence is unclear. Please rephrase for clarity.

A17) The sentence has been rephrased:

Three groups were formed with those cultivars having the highest J-index with each of the models. Unlike the previous study, in which groups were formed with all varieties that yielded higher J-index values with each model, in this case a condition was added to form the groups: it was not enough for the J-index values to be higher with each model, but the difference with the other two models had to be equal to or greater than one.

Q18) Figure 10: Separate images into panels in the figure and provide references to them in the figure caption. Sign axes in the middle and right images.

A18) Images have been separated into panels and references given in the figure caption. Legend to axes is now indicated.

Reviewer 3 Report

Comments and Suggestions for Authors

1. line 40: 'Vitis sylvestris' --> Scientific name in italics

2. line 77~91: If this is not a review paper, it seems unnecessary to indicate the formula in the introduction.

3. line 141: 'aspect ratio ' --> aspect ratio (AR)

4. Figures 2, 3, 4: The names of the varieties in the figures are not clearly visible. It would be better to make the names of the varieties more visible. If the names of these varieties are not necessarily included in the figures, it would be better to express the varieties of each group shown in the figures in a single table.

5. line 128 ~ "Three groups of cultivars were defined based on their ~~": In this study, three groups (Hebén, Chenin, and Regina models) were compared. In Table 1, EG is 'External Group'. However, in the PCA analysis in Figure 5, it was indicated as four groups. Of course, Group 1 means EG, but it would be better to revise it so that the notation and expression are consistent in the text. Or, it would be a good idea to use the group names in Table 1 as they are. That way, readers can read them easily and understand them easily.

6. Title: We need to consider whether the title of this study, 'A history of Vitis cultivars based on seed geometry', is appropriate. If it is not a review paper, I think it would be better to revise it a little so that it matches the core results of this study.

7. Citations: Please review and carefully revise it again so that it matches the format of the Plants journal, such as the abbreviation of the journal, the DOI notation, and the capitalization.

Author Response

Dear reviewer,

Thank you very much for your commentaries and corrections that have contributed to improve the quality of the article. The manuscript has been modified following your recommendations. Our answers (A) to specific questions (Q) are given below:

Q1. line 40: 'Vitis sylvestris' --> Scientific name in italics

A1. Done

Q2. line 77~91: If this is not a review paper, it seems unnecessary to indicate the formula in the introduction.

A2. We consider the formula are important to understand the meaning of the coefficients and the relative contribution of each of them to the diverse aspects of seed shape measurements that are later detailed in the results and discussion sections. 

Q3. line 141: 'aspect ratio ' --> aspect ratio (AR)

A3. Done

Q4. Figures 2, 3, 4: The names of the varieties in the figures are not clearly visible. It would be better to make the names of the varieties more visible. If the names of these varieties are not necessarily included in the figures, it would be better to express the varieties of each group shown in the figures in a single table.

A4. Names have been enlarged to be made visible.

Q5. line 128 ~ "Three groups of cultivars were defined based on their ~~": In this study, three groups (Hebén, Chenin, and Regina models) were compared. In Table 1, EG is 'External Group'. However, in the PCA analysis in Figure 5, it was indicated as four groups. Of course, Group 1 means EG, but it would be better to revise it so that the notation and expression are consistent in the text. Or, it would be a good idea to use the group names in Table 1 as they are. That way, readers can read them easily and understand them easily.

A5. Group names have been standardised throughout the text as follows: EG (External Group), Hebén, Chenin and Regina.

Q6. Title: We need to consider whether the title of this study, 'A history of Vitis cultivars based on seed geometry', is appropriate. If it is not a review paper, I think it would be better to revise it a little so that it matches the core results of this study.

A6. The title has been changed to:

Morphometric analysis reveals new data in the history of Vitis cultivars

Q7. Citations: Please review and carefully revise it again so that it matches the format of the Plants journal, such as the abbreviation of the journal, the DOI notation, and the capitalization.

A7. References have been reviewed.

Round 2

Reviewer 2 Report

Comments and Suggestions for Authors

The authors answered all the questions raised by the reviewer. The article can be accepted for publication.